# NINJ1 Regulates Platelet Activation and PANoptosis in Septic Disseminated Intravascular Coagulation

**DOI:** 10.3390/ijms24044168

**Published:** 2023-02-19

**Authors:** Xiaoli Zhou, Xiuxian Yu, Chengyu Wan, Fan Li, Yilan Wang, Kun Zhang, Lijuan Feng, Ao Wen, Jiangrong Deng, Shiyi Li, Guang Xin, Wen Huang

**Affiliations:** Laboratory of Ethnopharmacology, Tissue-Orientated Property of Chinese Medicine Key Laboratory of Sichuan Province, West China School of Medicine, West China Hospital, Sichuan University, Chengdu 610041, China

**Keywords:** NINJ1, platelet, plasma membrane rupture, sepsis, PANoptosis

## Abstract

Disseminated intravascular coagulation (DIC), which is closely related to platelet activation, is a key factor leading to high mortality in sepsis. The release of contents from plasma membrane rupture after platelet death further aggravates thrombosis. Nerve injury-induced protein 1 (NINJ1) is a cell membrane protein that mediates membrane disruption, a typical marker of cell death, through oligomerization. Nevertheless, whether NINJ1 is expressed in platelets and regulates the platelet function remains unclear. The aim of this study was to evaluate the expression of NINJ1 in human and murine platelets and elucidate the role of NINJ1 in platelets and septic DIC. In this study, NINJ1 blocking peptide (NINJ1_26–37_) was used to verify the effect of NINJ1 on platelets in vitro and in vivo. Platelet αIIbβ3 and P-selectin were detected by flow cytometry. Platelet aggregation was measured by turbidimetry. Platelet adhesion, spreading and NINJ1 oligomerization were examined by immunofluorescence. Cecal perforation-induced sepsis and FeCl_3_-induced thrombosis models were used to evaluate the role of NINJ1 in platelet, thrombus and DIC in vivo. We found that inhibition of NINJ1 alleviates platelet activation in vitro. The oligomerization of NINJ1 is verified in membrane-broken platelets, which is regulated by the PANoptosis pathway. In vivo studies demonstrate that inhibition of NINJ1 effectively reduces platelet activation and membrane disruption, thus suppressing platelet-cascade reaction and leading to anti-thrombosis and anti-DIC in sepsis. These data demonstrate that NINJ1 is critical in platelet activation and plasma membrane disruption, and inhibition of NINJ1 effectively reduces platelet-dependent thrombosis and DIC in sepsis. This is the first study to reveal the key role of NINJ1 in platelet and its related disorders.

## 1. Introduction

Sepsis is a life-threatening organ dysfunction caused by the host’s response to infection, and which has brought a heavy health and economic burden worldwide [1]. The Global Burden of Disease Study (GBA) recorded about 48.9 million cases of sepsis, with 11.0 million sepsis-related deaths reported, accounting for 19.7% of global disease deaths [2]. It is reported that inflammatory cytokine in sepsis stimulates the conversion of prothrombin to thrombin, which facilitates platelets activation to thrombosis [3]. The development of disseminated intravascular coagulation (DIC) during sepsis is recognized as a significant event associated with increased severity of sepsis, leading to higher mortality [4,5]. Platelets have emerged as critical cells implicated in the pathophysiology of sepsis, however, their role in exacerbating severe septic DIC has not been clearly defined. The benefits of anti-platelet therapy in sepsis also need to be further clarified.

Platelets are the smallest blood cells and are direct effectors of thrombotic diseases [6]. After platelet death and/or activation, DAMPs are released from the plasma membrane rupture into the circulation to promote a procoagulant phenotype, thereby promoting immunothrombosis [7,8,9]. This suggests that the platelet plasma membrane rupture is closely involved in the exacerbation of septic DIC.

NINJ1, a cell surface transmembrane protein, was first found to be expressed in the injured peripheral nervous system [10]. In recent years, NINJ1 has been found to be involved in a variety of pathophysiological processes, including nerve regeneration, nervous system inflammation, tumorigenesis, tissue homeostasis, etc. [11]. Here, whether NINJ1 is expressed in platelets is unknown. A recent study has shown that NINJ1 mediates plasma membrane disruption (PMR) downstream of various cell death processes in mouse macrophages through oligomerization [12]. However, whether NINJ1 can mediate the rupture of platelet plasma membrane through oligomerization and lead to the release of platelet contents to promote the process of septic DIC is also unclear.

In the present study, we demonstrate that inhibition of NINJ1 reduces platelet activation. Furthermore, we confirm that plasma membrane disruption of activated platelets is mediated by oligomerization of NINJ1. The results also reveal that inhibition of NINJ1 suppresses platelet PANoptosis and delays the progression of DIC in sepsis.

## 2. Results

### 2.1. Inhibition of NINJ1 Reduced the Degree of Agonist-Induced Platelet Activation In Vitro

Using immunofluorescence, we confirmed NINJ1 expression in platelets (Figure 1A). We also found that oligomerization of NINJ1 was observed in platelets with ruptured plasma membranes (Figure 1A). Next, we co-incubated the designed NINJ1 blocking peptide (NINJ1_26–37_) with platelets (Figure 1B), CCK8 results show that high concentrations of NINJ1_26–37_ had no significant cytotoxicity to platelets (Figure 1C).

Furthermore, we tested whether inhibition of NINJ1 could reduce platelet activation. Platelets pre-treated with NINJ1_26–37_ (5 or 10 μM) showed reduced αIIbβ3 activation (as measured by JON/A binding) compared with control (Figure 1D). Inhibition of NINJ1 decreased platelet α-granule secretion (as measured by P-selectin), whereas higher concentrations of NINJ1_26–37_ did not significantly inhibit α-granule secretion (Figure 1E). In addition, we found that inhibition of NINJ1 reduced the release of dense particles (as measured by ATP secretion) (Figure 1F). We also verified that inhibition of NINJ1 suppressed platelet aggregation induced by different agonists (thrombin (Thr), ADP, and U46619) as compared with solvent-treated controls (Figure 1F–I). These results suggest that inhibition of NINJ1 significantly prevents agonist-induced platelet activation.

### 2.2. Inhibition of NINJ1 Antagonizes the Platelet Integrin Outside-In Signaling In Vitro

First, we investigated the effect of NINJ1_26–37_ on platelet fibrin clot contraction to determine whether inhibition of NINJ1 could also modulate bidirectional signaling of transmembrane integrin αIIbβ3 by modulating “outside-in” signaling. We observed that coagulation retraction after thrombin stimulation was significantly inhibited in NINJ11_26–37_ pre-treated platelets (Figure 2A). Scanning electron microscope images showed that platelets in the control group formed dense clots, while inhibition of NINJ1 made the links between platelets looser (Figure 2B). Next, we observed platelet adhesion and spreading to fibrinogen at different times (0.5, 1.0, 1.5, or 2.0 h) (Figure 2C). The results show that inhibition of NINJ1 significantly decreased the number of adhered platelets at all four time points (Figure 2D). At 1.0 h, platelets treated with high concentration of NINJ11_26–37_ showed less spreading area than those in the model group, while inhibition of NINJ1 at 1.5 h and 2.0 h significantly lessened the spreading area (Figure 2E). Furthermore, using scanning electron microscopy, we observed that platelets pre-treated with NINJ1_26–37_ formed fewer pseudopodia (Figure 2F). These results suggest that inhibition of NINJ1 antagonized the platelet integrin outside-in signaling.

### 2.3. PANoptosis Pathway Regulates Platelet Plasma Membrane Disruption Related to NINJ1 Oligomerization by Calcium Overload

PANoptosis is defined as an inflammatory programmed cell death (PCD) pathway with plasma membrane rupture triggered by the synergistic effect of pyroptosis, apoptosis, and necroptosis [13,14]. We found that pretreatment with inhibition of AIM2 (AIM2-IN-3) and Caspase-8 (Ac-IETD-CHO) reduced the percentage of NINJ1 oligomerized platelets stimulated by thrombin. The intervention of Caspase-1 (Belancasan), Caspase-3 (Ac-DEVE-CHO), and MLKL inhibitors (Necrosulfon) alone had no significant difference. Simultaneous addition of the three inhibitors significantly reduced the percentage of platelet oligomerization in NINJ1 (Figure 3A–C). A large or sustained increase in Ca^2+^ concentration can trigger platelet death [15]. We found that oligomerization of NINJ1 was almost completely inhibited by the presence of Ca^2+^ chelators (BAPTA-AM) and thus stabilized platelet morphology. Intervention with NINJ1_26–37_ also inhibited the oligomerization of NINJ1 and reduced platelet activation (Figure 3A–C). These results suggest that oligomerization of NINJ1 mediates platelet plasma membrane disruption via PANoptosis pathways.

### 2.4. Inhibition of NINJ1 Reduced Arterial Thrombosis

Having evaluated the effect of NINJ1 on platelet function in vitro, we next determined the role of NINJ1 in thrombosis and hemostasis in vivo. The results show that, compared with the control mice, NINJ1_26–37_ (1 mg/kg) decreased thrombus formation (Figure 4A,B), and prolonged occlusion time (Figure 4C,D). We further examined its effect on hemostasis by measuring the time of tail-cut bleeding and found that the NINJ1_26–37_ treatment delayed hemostasis time (Figure 4E). Hematoxylin-eosin staining showed that treatment with NINJ1_26–37_ (1 mg/kg) reduced intravascular thrombus volume (Figure 5A), while Sirius red, EVG, and oil red staining showed that NINJ1_26–37_ intervention maintained thrombus stability (Figure 5B–D). These data suggest that inhibition of NINJ1 reduces arterial thrombosis.

### 2.5. Inhibition of NINJ1 Attenuates Platelet Activation and Inflammatory Cytokine Release in Sepsis

Next, cecal ligation and puncture (CLP) was used to establish a sepsis mouse model, and the role of NINJ1 in sepsis was investigated by tail vein injection of NINJ1_26–37_. A murine sepsis severity (MSS) score was used to assess the survival status of mice after modeling (Table 1). We found that NINJ1 inhibition improved the quality of life in septic mice and was significant at 24, 36, and 48 h (Figure 6A). The survival rate of septic mice treated with NINJ1_26–37_ was significantly higher than that of the model group (Figure 6B). The results show that treatment with NINJ1_26–37_ significantly reduced inflammatory infiltration and thrombosis in the lungs of septic mice (Figure 6C,D). We also examined the levels of inflammatory factors in serum samples of mice and found that the intervention with NINJ1_26–37_ reduced the secretion of TNF-α, IL-1β, and IL-6 (Figure 6E–G). Immunofluorescence was used to measure the expression of the platelet activation marker CD42b in lung tissue (Figure 6H), and the results showed that platelet activation was significantly reduced in the lungs of NINJ1_26–37_ treated septic mice (Figure 6I). Pretreatment with NINJ1_26–37_ reduced serum PF4 level (Figure 6J), as did platelet JON/A (Figure 6K) and CD62P (Figure 6L) expression. Additionally, NINJ1_26–37_ treatment also increased the platelet count in circulating blood (Figure 6M). At the same time, NINJ1_26–37_ treatment reduced the CLP-induced prolongation of PT (Figure 6N) and APTT (Figure 6O) and increased the CLP-induced reduction of Fbg (Figure 6P). These results suggest that inhibition of NINJ1 may alleviate platelet activation and inflammatory reaction in sepsis, thereby delaying the progression of DIC.

## 3. Discussion

In this study, we demonstrated for the first time that NINJ1 is expressed in platelets and is mainly distributed on platelet plasma membrane. We found that inhibition of NINJ1 effectively alleviated platelet activation and PANoptosis, thereby preventing thrombosis and progression of DIC in sepsis. These findings deepen our understanding of the relationship between sepsis and platelets and provide a novel strategy for the application of NINJ1 in the treatment of DIC in sepsis.

Many studies have revealed the key role of platelet activation in sepsis-induced DIC, and platelet activation is closely related to the progression of DIC [16,17]. Specifically, after platelet activation, α-granule, together with dense granule, is released from the plasma membrane to promote platelet cascade activation [18,19]. We found that inhibition of NINJ1 reduced the secretion of α-granule and dense granule in activated platelets. Integrin αIIbβ3 acts as a receptor for ligands that bridge platelets together [20], and ligand binding and integrin aggregation subsequently stimulate the “outside-in” signaling to drive platelet aggregation, spreading, and clot retraction [21,22]. We found that inhibition of NINJ1 reduced the activation of αIIbβ3 induced by thrombin, ADP and U46619. Inhibition of NINJ1 also decreased the number of adhering platelets and the area of platelet spreading. These results suggest that NINJ1 is involved in the functional regulation of platelets, including adhesion, aggregation, clot contraction and spreading.

It is well known that the plasma membrane is a direct barrier to the extracellular environment, and loss of plasma membrane integrity will undoubtedly result in cell death [23]. The oligomerization of NINJ1 has been reported to trigger damage of the plasma membrane in macrophages. In this study, we hypothesized that NINJ1 may mediate the rupture of the platelet plasma membrane through oligomerization, and that the release of platelet contents further promotes platelet activation and thrombosis. The results show that the expression of NINJ1 on the cell membrane was homogeneous in resting platelets, and that oligomerized NINJ1 existed on the broken membranes of platelets. Our study is consistent with the report from Kayagaki et al., that plasma membrane disruption after macrophage death is mediated by oligomerization of NINJ1 [24]. However, the relationship between oligomerization of NINJ1 in platelets and cell death types is still unclear.

PANoptosis is a mode of cell death characterized by plasma membrane rupture, which has key features that include pyroptosis, apoptosis, and necrosis [25]. Sepsis-induced DIC is characterized by high platelet activation and rapid death characterized by plasma membrane rupture, which is one of the main causes of high mortality in sepsis [26]. Calcium overload is characterized by excessive enrichment of Ca^2+^, leading to cell death with plasma membrane rupture as the outcome [27]. Therefore, we speculate that a calcium overload-related PANoptosis pathway occurs in platelets, resulting in a high cascade of platelet activation that further promotes thrombosis in septic DIC. Based on this, we used PANoptosis pathway inhibitors for validation in vitro. The results show that the oligomerization of NINJ1 on platelets was significantly reduced after AIM2 inhibition, which is consistent with the previous experiments by Lee et al., who confirmed that AIM2 plays an initiating regulatory role in PANoptosis [28]. Previous studies have revealed that Caspase-8 is critical for the formation of PANoptosis complex [29], and our study found that the addition of Caspase-8 inhibitor reduced the oligomerization of NINJ1. Moreover, inhibition of Caspase-1, Caspase-3 or MLKL alone did not significantly reduce NINJ1 oligomerization, but simultaneous treatment with the three inhibitors significantly reduced NINJ1 oligomerization. This is in line with previous studies in which inhibition of pyroptosis, apoptosis, or necroptosis alone did not inhibit the outcome of PANoptosis, and their co-treatment prevents the progression of PANoptosis [30,31]. Our studies further expand the understanding of the PANoptosis related to NINJ1 oligomerization in platelets.

Furthermore, we performed a FeCl_3_-induced carotid artery thrombosis model to explore the role of NINJ1 in thrombosis. The results show that intervention with NINJ1_26–37_ significantly reduced thrombus formation. The intervention with NINJ1_26–37_ prolonged the occlusion time of thrombosis. Therefore, we hypothesized that inhibition of NINJ1 in septic mice would reduce the occurrence of DIC. We used a cecal perforation model that was more consistent with the clinical characteristics of sepsis to assess the role of NINJ1 [32]. The results show that inhibition of NINJ1 could significantly reduce the activation of platelets and the formation of DIC in sepsis, thereby improving the survival rate of septic mice. Our study is consistent with that of Jennewein et al. who found that NINJ1_26–37_ improved 24 h survival and reduced plasma inflammatory factor levels in septic mice with perforated cecum [33]. Functional blockade of NINJ1 has been shown to protect diabetic endothelial cells both in vitro and in vivo [34]. NINJ1 deficiency also alleviated LPS/D-galactosamine-induced acute liver failure by reducing TNF-α-induced hepatocyte apoptosis [35]. Our study, for the first time, reveals the potential of NINJ1 as a therapeutic target for thrombosis and DIC in sepsis.

Overall, we have identified the critical role of NINJ1 in platelet activation and thrombosis in sepsis, and that this role is closely related to NINJ1-mediated platelet plasma membrane disruption. These findings may have important implications for understanding the relationship between platelets and sepsis. Our results indicate that inhibition of NINJ1 significantly delayed the progression of DIC in sepsis, which may provide new ideas for anti-platelet and anti-thrombotic therapy in patients with sepsis. The results advance the understanding of NINJ1 in platelet cell death and thrombosis, and also provide a novel therapeutic target for platelet-related diseases.

## 4. Materials and Methods

### 4.1. Animals

Eight-week-old SPF male C57BL/6 mice (Chengdu Dashuo Experimental Animal Co., Ltd., license number: SCXK chuan 2015-030), body weight 23 ± 2 g. The animal experiment and method were approved by the Ethics Committee of West China Hospital, Sichuan University (Ethics record number: 20220214019).

### 4.2. Materials

BAPTA-AM, AIM2-IN-3, Ac-IETD-CHO, Belancasan, Ac-DEVE-CHO and Necrosulfon were purchased from MCE. Thrombin and fibrillar type I collagen were purchased from Sigma. CD62P and αIIbβ3 integrin (Wug.E9, JON/A-PE, D200) were purchased from BIOHUB. NINJ1 antibody (sc-136295) was purchased from SCBIO. Goat anti-rabbit IgG-HRP (A0208) and goat anti-mouse IgG-HRP (A0216) were purchased from Beyotime Biotechnology.

### 4.3. Platelet Preparation and Count

Blood was collected from the hearts of mice anesthetized with pentobarbital and added to 3.8% citrate anticoagulant. Platelet-rich plasma (PRP) was obtained by centrifugation at 250 g for 6 min, followed by centrifugation of PRP at 600 g for 5 min to precipitate platelets. Platelets were resuspended in modified HEPES/Tyrode (H-T) buffer (136 mM NaCl, 0.4 mM Na_2_HPO_4_, 2.7 mM KCl, 12 mM NaHCO_3_, 0.1% glucose, 0.35% BSA, pH 7.4). Platelet cell counts were performed using a high-throughput multi-parameter cellular dynamic analysis system (PE/Opera Phenix Plus), with cell counts adjusted according to the needs of subsequent experiments.

### 4.4. Establishment of Experimental Septic Mouse Model

A clinically relevant mice model of sepsis was created by cecal ligation and puncture (CLP) [36]. At first, intraperitoneal anesthesia was administered (using 4% Nembutal) and the abdominal area was shaved and disinfected by applying an antiseptic solution. A 1 cm midline abdominal incision was made and the cecum was exposed. The distal half of the cecum was later ligated with a silk suture and was twice punctured through with a 22-gauge needle allowing the release of fecal material into the peritoneal cavity. Finally, the cecum was placed back into the peritoneal cavity and the incision was closed in two layers with a 3.0 suture. The severity of sepsis following CLP procedure was assessed based on murine sepsis severity (MSS) score [37]. Samples were collected from the mice at 6 h after CLP modeling.

### 4.5. CCK8

Platelets were resuspended and seeded at 5 × 10^3^ cells/well in a 96-well plate with 100 μL per well. Platelets were cultured with NINJ1_26–37_ at different concentrations (0, 1, 10, 100, 1000 μm) for 1 h at 37 °C. Additionally, 10 μL/well CCK8 solution was added to the culture plate for 2 h at 37 °C. The absorbance of each well at 450 nm was measured in a microplate reader.

### 4.6. Flow Cytometry

Mouse platelets were stimulated with thrombin (0.01 U/mL) in the presence or absence of NINJ1_26–37_ at various concentrations. PE-conjugated anti-mouse/rat P-selectin monoclonal antibody (Biolegend, 148306) and anti-mouse integrin αIIbβ3 (emfret, M023-2) monoclonal antibody were used for labeling. The mean fluorescence intensity (MFI) was measured by flow cytometry (BD Biosciences), data were analyzed using FlowJo v10 software.

### 4.7. Clot Retraction Assay

PRP (200 µL) was mixed with 5 µL of erythrocytes in a final volume of 1 mL prepared in a tabletop buffer in the presence or absence of various concentrations of NINJ1_26–37_. Thrombin (1 U/mL) was added to initiate clot formation, thrombosis was observed at room temperature for more than 2 h, and the volume of blood clot was quantified as a marker for retracting blood clot.

### 4.8. Platelet Aggregation

Rinsed platelets (2 × 10^8^/mL, 300 µL per sample) were resuspended in tabletop buffer containing fibrinogen and pre-incubated at 37 °C for 30 min (with or without NINJ1_26–37_) in platelet aggregation dishes. Thrombin, ADP, U46619, and collagen were used to start platelet aggregation under stirring conditions under the condition of 2 mM CaCl_2_. The changes in light transmission were monitored using a PAP-4 aggregator for 5 min.

### 4.9. Platelet Adhesion and Spreading Assays

Washed platelets at a density of 2 × 10^7^/mL were treated with NINJ1_26–37_ or control at various concentrations for 30 min, stimulated with 0.01 U/mL thrombin, and distributed on fibrinogen (50 µg/mL)-coated glassy bottom-well plates for 60 min. Unbound platelets were removed by washing with modified TyrodesHEPES buffer, and adherent platelets were fixed using 2% formaldehyde. Cells were stained with rhodamine-conjugated phalloidin and visualized using a multiparametric cell dynamic analysis system (PE/Opera Phenix Plus). Images were analyzed using ImageJ (NIH, Bethesda, MD, USA), and surface coverage and the amount of platelets adhering to fibrinogen were quantified.

### 4.10. Scanning Electron Microscope

The samples were fixed with 2.5% glutaraldehyde at 4 °C overnight, washed five times with PBS the next day, and further subjected to gradient dehydration with 30–100% ethanol and critical point drying in a Leica EM CPD300. The samples were plated on an Emitech sc7320 sputter coater and visualized with a ZEISS EVO10 scanning electron microscope.

### 4.11. Tail Bleeding Assay

After the mice were anesthetized with pentobarbital, the tail was truncated 2 mm from the tip of the tail with a razor blade. The tail was immediately placed in a centrifuge tube containing preheated 0.9% NaCl to continuously monitor bleeding until it stopped completely, and the time was recorded. In the case of continuous bleeding, the experiment was stopped after 600 s and the bleeding was stopped by pressure.

### 4.12. Vessel Occlusion after FeCl_3_-Injury of the Carotid Artery

Calcein green labeled platelets (2.5 × 10^9^ platelets/kg) were injected into the tail vein of mice, anesthetized with 1% pentobarbital, and then the common carotid artery was carefully exposed. The common carotid artery was kept for local application with 5% ferric chloride (FeCl_3_) saturated filter paper (0.5 × 1.5 mm) for 1 min. Thrombosis in injured carotid vessels was monitored using Nikon A1RMP+ two-photon microscopy, and occlusion times exceeding 30 min were recorded as 30 min. The time an occlusive thrombus forms is considered to be the time when blood has stopped flowing completely. One minute is the required time.

### 4.13. Statistical Analysis

GraphPad Prism software version 8 (GraphPad Software, San Diego, CA, USA) was used for statistical analysis. Statistical significance was assessed by one-way ANOVA or two-way ANOVA followed by Tukey’s multiple comparison test (for normally distributed data). Kruskal–Wallis test followed by Dunn’s multiple comparison test (for non-normally distributed data). All data are presented as mean ± standard error (SEM), *p* < 0.05 was considered statistically significant.

## Figures and Tables

**Figure 1 ijms-24-04168-f001:**
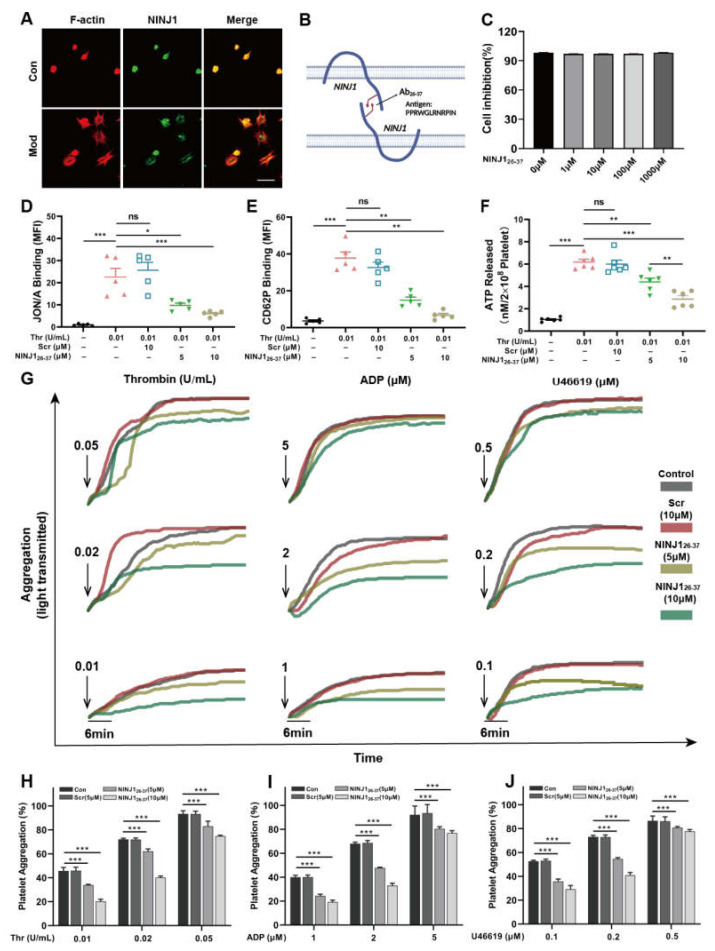
Inhibition of NINJ1 reduces the degree of agonist-induced platelet activation. (**A**) Washed platelets adhere and spread to fibrinogen coated wells by stimulation with thrombin for 1.5 h at 37 °C, model group pre-stimulated with thrombin (0.01 U/mL). After being fixed and stained, the platelets were observed with a fluorescence microscope. Scale bar represents 5 µm. (**B**) Design of NINJ1 inhibitory peptides. (**C**) Cytotoxicity of NINJ1 inhibitory peptide. (**D**) Effect of inhibition of NINJ1 on integrin αIIbβ3 activation, (**E**) P-selectin exposure, and (**F**) ATP secretion in thrombin (0.01 U/mL) stimulated-platelets. Values are mean ± SEM, *n* = 5. ns = *p* > 0.05, * *p* < 0.05, ** *p* < 0.01, *** *p* < 0.001. (**G**) Washed platelets in the presence of fibrinogen pre-treated with vehicle, Scr (10 μM) or NINJ1_26–37_ (5 or 10 μM) and stimulated with (**H**) thrombin (0.01, 0.02, or 0.05 U/mL), The different colors of the groups information shown on the right. (**I**) ADP (1, 2, or 5 μM), and (**J**) U46619 (0.1, 0.2, or 0.5 μM). The groups information represented by the different color bars is shown at the top of the figure. Results are expressed as the percentage change in light transmission with respect to the blank (buffer without platelets), set at 100%. Values are mean ± SEM, *n* = 5. * *p* < 0.05, ** *p* < 0.01, *** *p* < 0.001. One-way ANOVA followed by Tukey’s multiple comparisons test.

**Figure 2 ijms-24-04168-f002:**
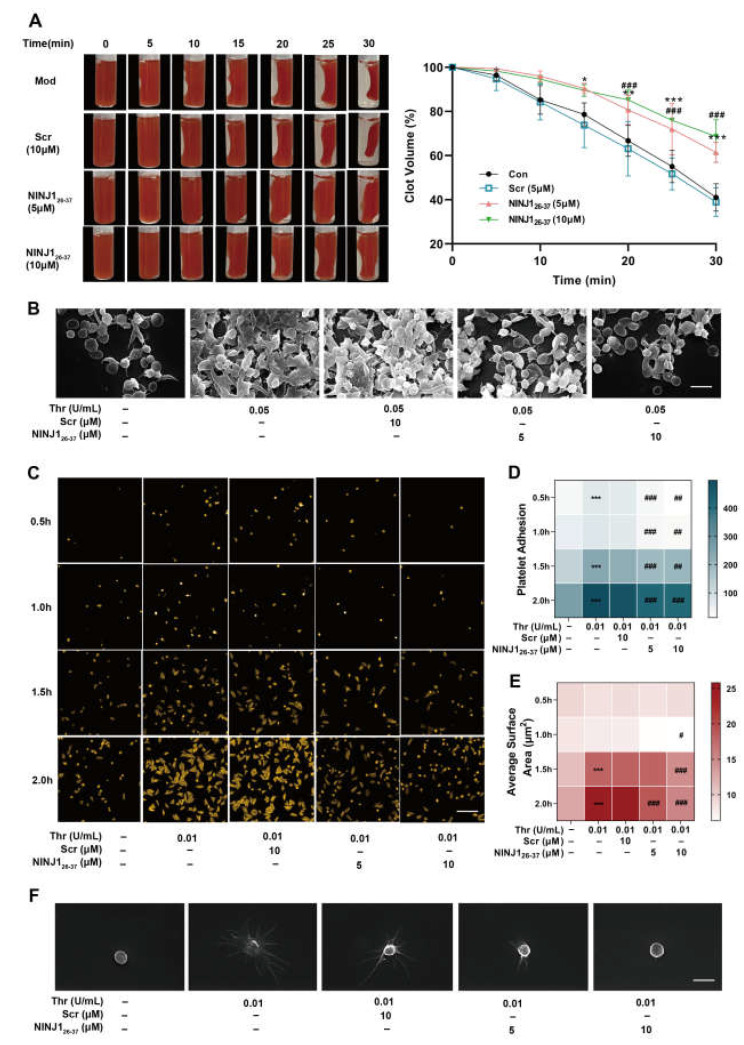
Inhibition of NINJ1 weakens platelet integrin signaling. (**A**) Clot retraction was measured for 30 min in washed platelets in fibrinogen, supplemented with RBC, after adding 1.0 U/mL thrombin in the presence of a vehicle, Scr (10 μM), or NINJ1_26–37_ (5 or 10 μM). The left panels show representative images at different times. The right panel shows the quantification of clot size. Values are mean ± SEM, *n* = 4. Two-way ANOVA with Tukey’s multiple comparison test. (**B**) Scanning electron microscopy was performed to assess the extent of platelet aggregation and the ultrastructure. Scale bar represents 2 µm. (**C**) Washed platelets adhere to and spread on fibrinogen with thrombin (0.01 U/mL) stimulation at 37 °C for 0.5 to 2 h. Once fixed and stained, the platelets were observed with a fluorescence microscope. Images were acquired, and the spreading area of single platelets was measured using ImageJ software, with pixel number as the unit of size. Scale bar represents 10 µm. (**D**) Number of adhesion and (**E**) surface areas of single platelets from seven randomly selected fields of three different tests. * *p* < 0.05, ** *p* < 0.01, *** *p* < 0.001 vs. resting platelets; # *p* < 0.05, ## *p* < 0.01, ### *p* < 0.001 vs. activated platelets. (**F**) Scanning electron microscopy was performed to assess the extent of platelet spread and the ultrastructure. Representative scanning electron microscopy images of platelets treated with vehicle, Scr (10 µM), NINJ1_26–37_ (5 or 10 µM), and then stimulated with thrombin. Scale bar represents 1 µm.

**Figure 3 ijms-24-04168-f003:**
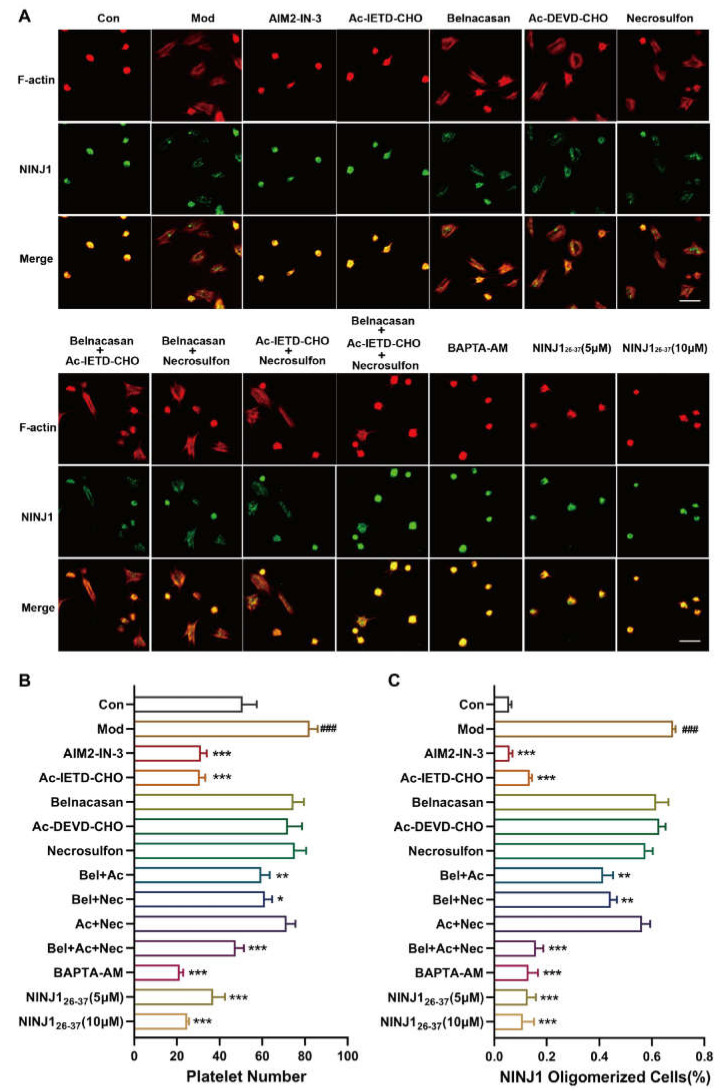
Oligomerization of NINJ1 in membrane-ruptured platelets is regulated by the PANoptosis pathway. (**A**) Washed platelets incubated with different inhibitors (AIM2-IN-3, Ac-IETD-CHO, Belnacasan, Ac-DEVD-CHO, Necrosulfon, BAPTA-AM, NINJ1_26–37_) adhered to and spread on fibrinogen with thrombin (0.1 U/mL) for 1.5 h at 37 °C. Compared with the control group, the other groups were pretreated with thrombin (0.01 U/mL). Once fixed and stained, the platelets were observed with a fluorescence microscope. Scale bar represents 5 µm. (**B**) Number of adhesion and (**C**) oligomerization ratio of NINJ1 from five randomly selected fields of three different tests. Values are mean ± SEM, * *p* < 0.05, ** *p* < 0.01, *** *p* < 0.001 vs. resting platelets; ### *p* < 0.001 vs. activated platelets.

**Figure 4 ijms-24-04168-f004:**
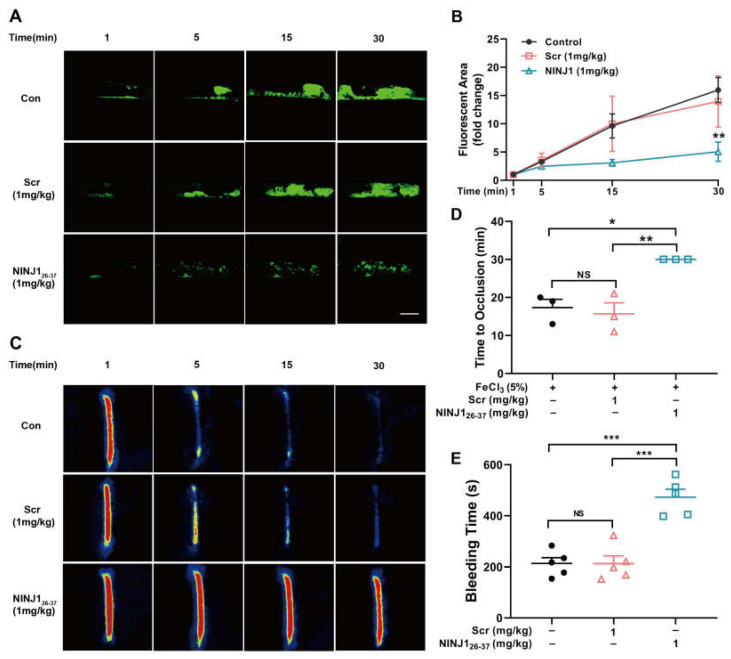
Inhibition of NINJ1 reduced FeCl3-induced arterial thrombosis. (**A**) The panel shows representative microphotographs of carotid artery thrombus (5% FeCl_3_ injury) as visualized by upright intravital microscopy in male mice. Platelets were labeled in vitro with calcein (*n* = 3). (**B**) The rate of thrombus growth was calculated by dividing the area of the thrombus at different times (5, 15, or 30 min) by the area of the same thrombus at the time (1 min). Values are mean ± SEM, * *p* < 0.05, ** *p* < 0.01, *** *p* < 0.001. Two-way ANOVA with Tukey’s multiple comparison test. (**C**) The panel shows a representative pseudo-color photograph of blood flow of carotid artery thrombus (5% FeCl_3_ injury) observed with a laser speckle contrast imaging. The red color represents the high blood perfusion signal. Yellow represents a low blood flow perfusion signal. (**D**) The panel shows the time to occlusion (*n* = 3). (**E**) Tail bleeding time was measured after the tip of the tail was removed 2 mm, values are mean ± SEM, *n* = 5. NS = *p* > 0.05, * *p* < 0.05, ** *p* < 0.01, *** *p* < 0.001. One-way ANOVA with Tukey’s multiple comparisons.

**Figure 5 ijms-24-04168-f005:**
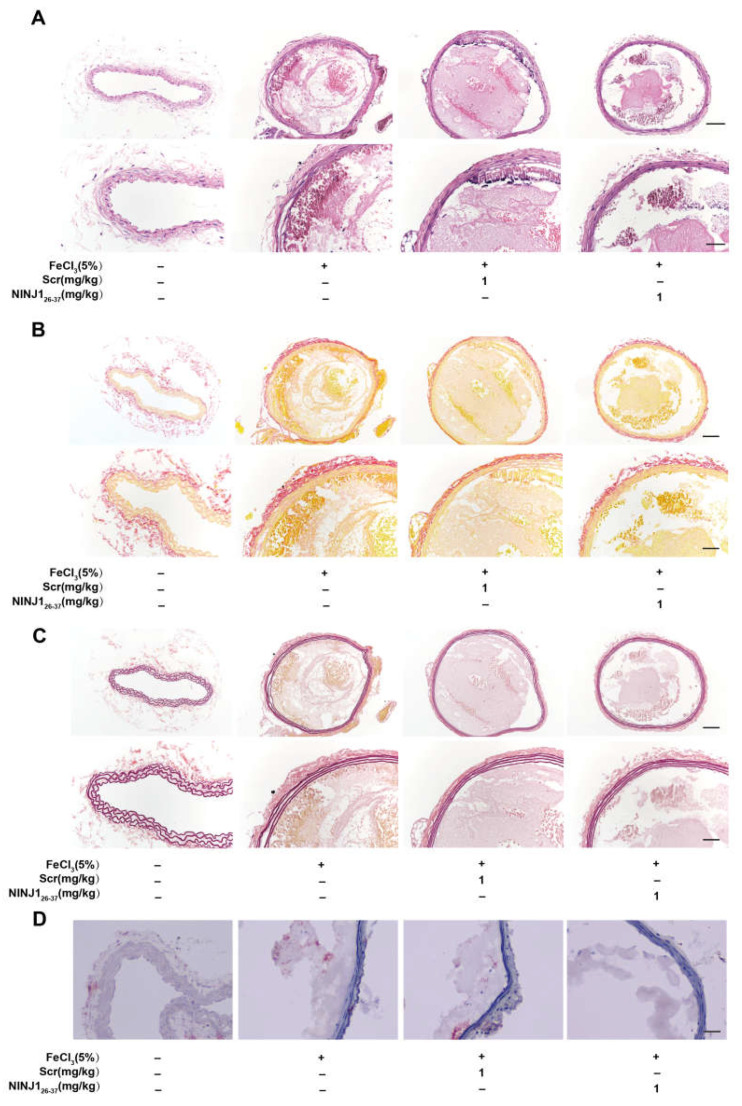
Inhibition of NINJ1 reduced thrombosis without affecting its stability. The panel shows images of carotid thrombosis (5% FeCl_3_ injury) with representative (**A**) hematoxylin-eosin staining, (**B**) Sirian red, (**C**) EVG (scale bar represents 100 μm or 50 μm), and oil (**D**) red staining, scale bar represents 100 μm.

**Figure 6 ijms-24-04168-f006:**
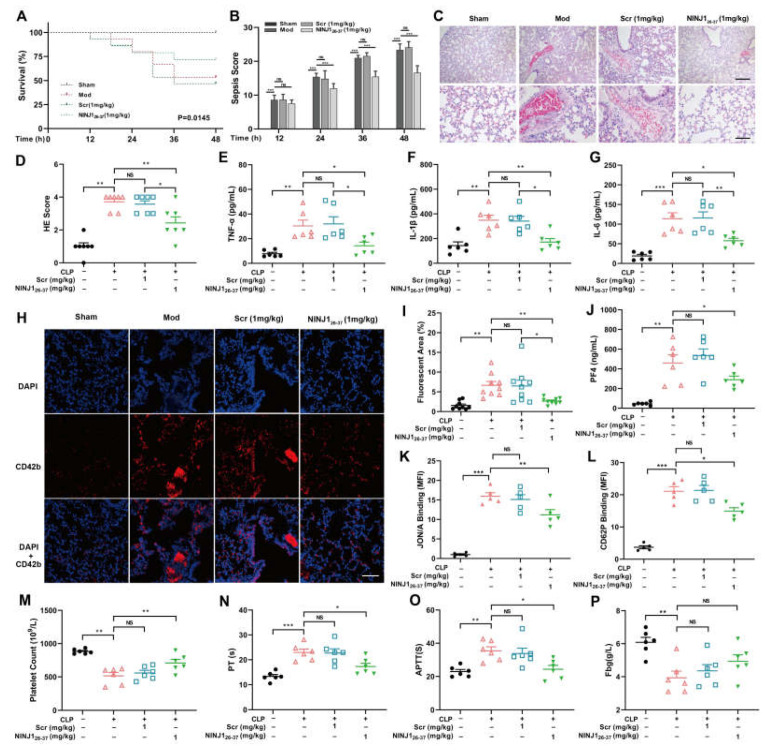
Inhibition of NINJ1 attenuates platelet activation and inflammatory reaction in sepsis. (**A**) In the CLP-induced sepsis murine model, mice with sepsis were evaluated with MSS scores at different times (12, 24, 36, or 48 h). Values are mean ± SEM, *n* = 15. * *p* < 0.05, ** *p* < 0.01, *** *p* < 0.001. Two-way ANOVA with Tukey’s multiple comparison test. (**B**) A 48-h log-rank (Mantel–Cox) test in the CLP-induced sepsis murine model was performed (*n* = 15). (**C**) Lung tissues of mice with CLP-induced sepsis were collected for HE staining and (**D**) lung injury score. Values are mean ± SEM, *n* = 7. Scatter diagrams showing plasma levels of TNF-α (**E**), IL-1β (**F**) and IL-6 (**G**) in the CLP-induced sepsis murine model. Values are mean ± SEM, *n* = 6. (**H**) Lung tissues of mice with CLP-induced sepsis were collected, and, once fixed and stained with CD42b and DAPI, were observed with a fluorescence microscope. Scale bar represents 50 µm. (**I**) Fluorescent area of CD42b from nine randomly selected fields of three different tests. (**J**) Scatter diagrams showing plasma levels of PF4 in the CLP-induced sepsis murine model. Values are mean ± SEM, *n* = 6. (**K**) Effect of inhibition of NINJ1 on integrin αIIbβ3 activation and (**L**) P-selectin exposure in CLP-induced sepsis murine model platelets, *n* = 5. Blood was collected from mice for measuring the (**M**) platelet count, (**N**) PT, (**O**) APTT and (**P**) Pbg, *n* = 6. NS = *p* > 0.05, * *p* < 0.05, ** *p* < 0.01, *** *p* < 0.001. One-way ANOVA with Tukey’s multiple comparisons.

**Table 1 ijms-24-04168-t001:** A murine sepsis severity (MSS) score was used to assess disease severity in a CLP–sepsis model.

Variable	Score and description
Appearance	0—Coat is smooth
	1—Patches of hair piloerected
	2—Majority of back is piloerected
	3—Piloerection may or may not be present, mouse appears “puffy”
	4—Piloerection may or may not be present, mouse appears emaciated
Level of consciousness	0—Mouse is active
	1—Mouse is active but avoids standing upright
	2—Mouse activity is noticeably slowed. The mouse is still ambulant.
	3—Activity is impaired. Mouse only moves when provoked, movements have a tremor
	4—Activity severely impaired. Mouse remains stationary when provoked, with possible tremor
Activity	0—Normal amount of activity. Mouse is any of: eating, drinking, climbing, running, fighting
	1—Slightly suppressed activity. Mouse is moving around bottom of cage
	2—Suppressed activity. Mouse is stationary with occasional investigative movements
	3—No activity. Mouse is stationary
	4—No activity. Mouse experiencing tremors, particularly in the hind legs
Response to stimulus	0—Mouse responds immediately to auditory stimulus or touch
	1—Slow or no response to auditory stimulus; strong response to touch (moves to escape)
	2—No response to auditory stimulus; moderate response to touch (moves a few steps)
	3—No response to auditory stimulus; mild response to touch (no locomotion)
	4—No response to auditory stimulus. Little or no response to touch. Cannot right itself if pushed over
Eyes	0—Open
	1—Eyes not fully open, possibly with secretions
	2—Eyes at least half closed, possibly with secretions
	3—Eyes half closed or more, possibly with secretions
	4—Eyes closed or milky
Respiration rate	0—Normal, rapid mouse respiration
	1—Slightly decreased respiration (rate not quantifiable by eye)
	2—Moderately reduced respiration (rate at the upper range of quantifying by eye)
	3—Severely reduced respiration (rate easily countable by eye, 0.5 s between breaths)
	4—Extremely reduced respiration (>1 s between breaths)
Respiration quality	0—Normal
	1—Brief periods of labored breathing
	2—Labored, no gasping
	3—Labored with intermittent gasps
	4—Gasping

## Data Availability

Data are contained within the article.

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
