# Peer review of "NINJ1 Regulates Platelet Activation and PANoptosis in Septic Disseminated Intravascular Coagulation"

_ijms, 2023, doi:10.3390/ijms24044168_

Round 1
Reviewer 1 Report
General Comments
In their manuscript, Zhou et al. demonstrate that NinJ1 contributes to platelet activation, which likely promotes inflammatory disorders, including sepsis and disseminated intravascular coagulopathy. Although the manuscript requires some language and spelling alterations, the article is overall well-written and with clear figures depiction and appropriate statistical analysis. To improve the novelty and impact of their manuscript, the authors should directly assess platelet function and markers of disseminated intravascular coagulopathy in their sepsis model (see suggestions in Major Comments). Such work is important when demonstrating the importance of platelet NinJ1 to inflammatory diseases, especially as their inhibitory peptide will block NinJ1 functions in other associated cells. There are also serious concerns over the duplication of images, and the lack of acknowledgment/discussion of similar work that must be addressed.
Overall, the manuscript has novelty and is of interest to the platelet/inflammation research community.
Major Comments
1. Image duplication: The “con” and “mod” images from Figures 1A and 3A are identical. This must be corrected to avoid accusations of data manipulation.
2. The authors do not cite and discuss the work of Jennewein et al., (2015), which describes the contribution of Ninjurin1 to inflammation in the same cecal ligation model. This is a serious omission and the authors need to discuss how their own experiments (e.g., survival outcomes) compare to previously published work.
3. For their sepsis model, the authors state that Ninjurin-1 inhibition attenuates platelet activation. Although CD42b is a platelet marker, it does not distinguish if platelets are activated and there are no other assays that look specifically at platelet activation. This is an important consideration when their NINJ1 inhibitory peptide is known to affect other cells involved in thrombosis and sepsis/inflammation (Jennewein, 2015). The authors should isolate platelets from septic mice with/without NINJ1 treatment and assess some platelet functions (e.g., CD62p, JON/A, platelet spreading) to confirm that NINJ1 affects platelets in their model of sepsis. The authors should also test for soluble markers of platelet activation (e.g., platelet factor 4) and DIC in plasma to justify their statements.
4. The author’s data suggest a slight recovery in platelet counts in their sepsis models following NINJ1 inhibition. To check if this is due to reduced platelet clearance or consumption, the authors should track platelet lifespan through methods such as the in vivo administration of biotin.
5. Line 265-266: In their discussion, the authors state they performed FeCl3-induced carotid arterial thrombosis to explore the role of NINJ1 in sepsis-induced DIC. However, these experiments were conducted in healthy mice. This needs clarification.
Minor Comments
- Technically, NINJ1 has previously been described in the platelet proteome (Burkhart et al. 2012). The authors should adjust their language to show they are instead the first to validate NINJ1 expression in platelets and show it contributes to platelet function.
- The authors need to first define their abbreviated NinJ1 as Nerve injury-induced protein 1.
- Figure 1A: the authors state the scale bar is 80 um. Given the diameter of a platelet is 2-3 um, this needs clarification.
- Figure 1A, 3A: the authors do not explain their two conditions “Con” and “Mod”.
- Figure 1D: the authors need to explain/define that “Scr” refers to a scrambled peptide control.
- The authors should include their CCK8 assay in the methods.
- Line 77: The authors likely mean U46619, not U46629
- Line 75: Can the authors please refer to the ATP-rich bodies in platelets as “dense granules” so as not to confuse the reader.
- Figure 3C: The y-axis states %, but the numbers provided are a ratio. Please select either a percentage or ratio when discussing oligomerized cells.
- Figure 4: The authors state in their methods that Calcein-labeled platelets were injected into the tail vein of recipient animals, but in the legend of Figure 4, they state platelets were labeled in vivo. The authors need to provide more consistency and clarity in their methods.
- The authors include in their methods how platelet counts were obtained.
- In their discussion, the authors state that NINJ1 is “mainly distributed on the platelet plasma membrane”. Unless the platelets were not permeabilized in their immunostaining protocol (which is not included in their methods), the authors need to either remove this comment or provide experiments (e.g., fractionation, Z stacks) that show platelet NINJ1 is on/in the plasma membrane.
Author Response
Dear reviewer:
Thank you for your constructive comments on our manuscript. We have carefully considered the suggestion and make some changes. We have tried our best to improve and made some changes in the manuscript. Revision notes, point-to-point, are given as follows:
Major Comments
1.Image duplication: The “con” and “mod” images from Figures 1A and 3A are identical. This must be corrected to avoid accusations of data manipulation.
Thanks for your correction, this problem is indeed due to an oversight in our work. In fact, Figures 1A and 3A are from the same experiment and because we wanted to clarify the expression of NINJ1 in platelets at the beginning of the article and present our findings on NINJ1 oligomerisation, we placed the control and model group images from Figure 3A again in Figure 1A. We really did not consider that this might raise a question, so we reselected the control and model pictures in Figure 1A.
2.The authors do not cite and discuss the work of Jennewein et al., (2015), which describes the contribution of Ninjurin1 to inflammation in the same cecal ligation model. This is a serious omission and the authors need to discuss how their own experiments (e.g., survival outcomes) compare to previously published work.
Thanks to your suggestion, we have carefully referenced Jennewein et al.'s, (2015) study and added a discussion of its similarities with our study to the article.
- For their sepsis model, the authors state that Ninjurin-1 inhibition attenuates platelet activation. Although CD42b is a platelet marker, it does not distinguish if platelets are activated and there are no other assays that look specifically at platelet activation. This is an important consideration when their NINJ1 inhibitory peptide is known to affect other cells involved in thrombosis and sepsis/inflammation (Jennewein, 2015). The authors should isolate platelets from septic mice with/without NINJ1 treatment and assess some platelet functions (e.g., CD62p, JON/A, platelet spreading) to confirm that NINJ1 affects platelets in their model of sepsis. The authors should also test for soluble markers of platelet activation (e.g., platelet factor 4) and DIC in plasma to justify their statements.
Thanks for your valuable suggestions. We examined the expression of JON/A and CD62p on platelets from septic mice using flow cytometry, and plasma from mice was collected for PF4 ELISA test and coagulation function test to further clarify the role of NINJ1 in septic DIC. The results are presented in Figure 6 and described in the text.
- The author’s data suggest a slight recovery in platelet counts in their sepsis models following NINJ1 inhibition. To check if this is due to reduced platelet clearance or consumption, the authors should track platelet lifespan through methods such as the in vivo administration of biotin.
Thank you for your suggestion. Biotin experiment is indeed needed. However, we are not familiar with this experiment at present and cannot supplement it for the time being. If possible, we would like to use platelet count test together with coagulation function test as an index to evaluate DIC.
We will apply this technique in future studies with a view to providing a comprehensive explanation for the phenomenon of platelet count changes. Thank you again for your advice.
- Line 265-266: In their discussion, the authors state they performed FeCl3-induced carotid arterial thrombosis to explore the role of NINJ1 in sepsis-induced DIC. However, these experiments were conducted in healthy mice. This needs clarification.
We thank the expert for the correction, our description was indeed inappropriate. Our aim in using the FeCl3 model was to first explore the possible role of NINJ1 in arterial thrombosis and later to conduct further studies in a septic disease model. We have revised our description in the text to avoid ambiguity.
Minor Comments
- Technically, NINJ1 has previously been described in the platelet proteome (Burkhart et al. 2012). The authors should adjust their language to show they are instead the first to validate NINJ1 expression in platelets and show it contributes to platelet function.
Thank you for your comments, our previous research into the background of the literature did contain omissions. We have read this literature and corrected our previous descriptions.
- The authors need to first define their abbreviated NinJ1 as Nerve injury-induced protein 1.
We have defined NINJ1 as nerve injury-induced protein 1 at the time of its first appearance.
- Figure 1A: the authors state the scale bar is 80 um. Given the diameter of a platelet is 2-3 um, this needs clarification.
When writing the manuscript, we referred to other literature for the description of the scale and copied it into our article, but due to an oversight on our part, we omitted to correct the value to such an extent that the error occurred, and we have revised the scale to 5 μm in the figure legend.
- Figure 1A, 3A: the authors do not explain their two conditions “Con” and “Mod”.
We have supplemented the description of these two conditions in the figure legends.
- Figure 1D: the authors need to explain/define that “Scr” refers to a scrambled peptide control.
Line 70:We supplemented the definition of “Scr” in 2.1.
- The authors should include their CCK8 assay in the methods.
We have added the description of CCK8 to our methods.
- Line 77: The authors likely mean U46619, not U46619.
We have changed U46629 to U46619.
- Line 75: Can the authors please refer to the ATP-rich bodies in platelets as “dense granules” so as not to confuse the reader.
Thanks to your suggestion, we have revised the description in the text.
- Figure 3C: The y-axis states %, but the numbers provided are a ratio. Please select either a percentage or ratio when discussing oligomerized cells.
We have modified the y-axis of Figure 3C.
- Figure 4: The authors state in their methods that Calcein-labeled platelets were injected into the tail vein of recipient animals, but in the legend of Figure 4, they state platelets were labeled in vivo. The authors need to provide more consistency and clarity in their methods.
The use of calcium yellow-green labelled platelets was indeed performed in vitro and later injected into the body for subsequent experiments. We have removed this inaccurate description from the figure notes to avoid misleading readers.
- The authors include in their methods how platelet counts were obtained.
We have added a description of how to perform a platelet count to the methods 4.3.
- In their discussion, the authors state that NINJ1 is “mainly distributed on the platelet plasma membrane”. Unless the platelets were not permeabilized in their immunostaining protocol (which is not included in their methods), the authors need to either remove this comment or provide experiments (e.g., fractionation, Z stacks) that show platelet NINJ1 is on/in the plasma.
Thanks for your valuable advice. Although we did not perform permeabilization when performing the immunofluorescence experiments, we are unable to provide additional validation (e.g., fractionation, Z stacks) at this time. To ensure rigour in our conclusions, we have removed this state.
In addition, we added in the methods that we did not perform a permeabilization step in platelet adhesion and spreading assays.

Reviewer 2 Report
The article entitled “NINJ1 regulates platelet activation and PANoptosis in septic disseminated intravascular coagulation” focuses on a new marker of platelet activation in disseminated intravascular coagulation (DIC). DIC is a severe and life-threatening complication in infectious disesaes, in hematological neoplasms, and obstetric complications and an article that adresses this topic is well welcome. The study project is well-conducted and the laboratory and experimental evaluation of the functionla role of NINJ1 is appropriate. At the moment, there are not studies about the thrombotic role of NINJ1 protein in human. Therefore, I think that this article is suitable for publication in its current version.
Author Response
Dear Reviewer,
Thank you for reviewing our manuscript in the midst of your busy schedule. Your approval is of paramount importance to us and will certainly be a great encouragement to our future research.
Once again, we would like to thank you most sincerely.

Round 2
Reviewer 1 Report
The resubmitted manuscript by Zhou et al., adequately addresses my previous comments and concerns. Their work is clear, well-presented, and in my opinion ready for publication. I appreciate the author's detail within their response letter and congratulate them on their work.